# Who said what? Speaker Identification from Anonymous Minutes of Meetings

**Daniel Holmer, Lars Ahrenberg, Julius Monsen, Arne Jönsson**
Department of Computer and Information Science
Linköping University, Sweden
daniel.holmer|lars.ahrenberg|julius.monsen|arne.jonsson@liu.se

**Mikael Apel**
Sveriges Riksbank
mikael.apel@riksbank.se

**Marianna Blix Grimaldi**
The Swedish National Debt Office
marianna.blixgrimaldi@riksgalden.se

## Abstract

We study the performance of machine learning techniques to the problem of identifying speakers at meetings from anonymous minutes issued afterwards. The data comes from board meetings of Sveriges Riksbank (Sweden's Central Bank). The data is split in two ways, one where each reported contribution to the discussion is treated as a data point, and another where all contributions from a single speaker have been aggregated. Using interpretable models we find that lexical features and topic models generated from speeches held by the board members outside of board meetings are good predictors of speaker identity. Combining topic models with other features gives prediction accuracies close to 80% on aggregated data, though there is still a sizeable gap in performance compared to a not easily interpreted BERT-based transformer model that we offer as a benchmark.

## 1 Introduction

Attributing a text or a part thereof to an agent is a well-established sub-field of computational linguistics. Apart from the traditional task of author attribution, it has also been applied in social media studies, to the identification of speakers in fiction dialogues, and for detection of plagiarism. In this work, we study a new but related problem: identifying speakers at meetings from anonymous minutes issued afterwards.

The data at hand are minutes, in Swedish, from the monetary policy meetings of the Riksbank's Executive board. The main monetary policy objective is to keep inflation low and stable, close to the target of 2 percent. The key issue at the meetings is to decide on the policy rate, and, since the global financial crisis in 2007-2009, on purchases of financial assets. Minutes from meetings like these are not only common for central banks but also, for instance, corporates, c.f. (Agarwala et al., 2022; Schwartz-Ziv and Weisbach, 2013).

Until June 2007 the minutes of the Swedish Riksbank's monetary policy meetings gave an anonymised account of the deliberations. Since then, however, the identity of a board member is revealed in the minutes so that it is possible to know which member expressed which opinion during the meeting. This change towards increased transparency is of great interest to researchers on economic policy-making and there is a growing literature in this area (Hansen et al., 2018). It could potentially affect board members' incentives and behaviour in different ways, not least because the minutes are published only around two weeks after a meeting.

Following the theoretical literature increased transparency can have different effects. It can make agents prepare more thoroughly – a disciplinary effect (Holmström, 1999). It can also make agents behave differently due to career concerns, either by making them less inclined to oppose to the majority view – a herding, or conformism, mechanism – or by making them instead want to distinguish themselves more from others – an anti-herding or exaggeration mechanism. It may also make agents more committed to stick to a specific opinion once they have expressed it and less willing to change their mind, even if circumstances change (Falk and Zimmermann, 2018).

| Swedish | English translation |
|---|---|
| Vice riksbankschef A inledde diskussionen med att uttrycka sitt stöd för det B sade om behovet av att ha en bredare ansats när man analyserar skälen till den låga inflationen. Här är det, menade han, viktigt att ta hänsyn till både efterfråge- och utbudsfaktorer. Att fokus varit ensidigt kan möjligen vara förståeligt i länder som befunnit sig i krisens epicentrum, fortsatte A. Där har stora negativa effekter på produktion, sysselsättning och arbetslöshet helt dominerat både debatten och den ekonomiska politikens inriktning. | Deputy Governor A started the discussion by expressing his support for what B had said about the need for a broader approach when analysing the reasons for the low inflation rate Here, he said, it is important to consider factors of both demand and supply. That there has been a one-sided focus may be understandable in countries that have been at the epicentre of the crisis, A continued. There great negative effects on production and employment have dominated both the debate and the direction of economic policies. |

Table 1: Extract from a contribution.

Here we are not concerned with transparency effects as such, rather we want to find out what features and methods would enable us to trace the behaviour of individual members when conditions are changed, from a state where views, but not identities, are reported in the minutes, to a state where both identities and views are revealed. The study can be seen as a first contribution to the development of automatic tools that can support transparency studies by analysing minutes of meetings created under different conditions.

In this study we investigate the problem of predicting agent identities under a supervised condition, using minutes from the period September 2007 to April 2018 for experiments. During this period the board has had six members at any given time, but as members have limited periods of service, altogether twelve people have served on the board. We are looking for features of the board members that can be assumed to be relatively stable over time, and so be used for identification. The study is thus an experiment in de-anonymisation, which has been defined as a reverse engineering process in which de-identified data are cross-referenced with other data sources to re-identify the personally identifiable information[1]. The data to be re-identified are participants' contributions to the discussions preceding the vote on policy rate as they are reported in the minutes. The primary data used for cross-referencing are speeches made by the members to private and public audiences outside of board meetings. Both the minutes of the meetings and the speeches are pub-

licly available on the Riksbank's website.

The minutes are compiled by a secretary who has access to recordings of the meeting. Discussions and decisions are reported in detail using a formal writing style where sentences are well-formed and punctuation formal. For an example, see Table 1. During a meeting a member may make several contributions and the start of a new contribution is usually marked in the minutes by a reference including the title and full name of the member. A contribution can be short, only a few sentences, but sometimes as long as several paragraphs. The minutes may sometimes partly be based on written notes provided by members but we do not know to what extent this happens nor how much editing is done.

The aims of the study are three-fold: 1) to compare the performance of several machine learning methods on this task, all of which have been successfully applied to attribution tasks in the past; 2) to identify features of members and their contributions that can aid de-anonymisation; 3) to establish a benchmark for what can likely be achieved on anonymised minutes under an unsupervised condition.

The methods investigated are:

- Burrows' Delta (Burrows, 2002)
- A Support Vector Machine (SVM)
- A Multi-layer Perceptron (MLP)
- Two ensemble methods of SVMs and MLPs
- A Swedish BERT model (Malmsten et al., 2020) fine-tuned for sequence classification

The paper is organised as follows. In section 2 we report related work. In section 3 we describe our data and the preprocessing we have applied. In

---

[1]https://codata.org/rdm-glossary/de-anonymization/

section 4 we describe the features we have used in the study and in particular the topic model we have used. Section 5 reports our experiments and the final sections discuss our results and report our conclusions.

## 2 Related work

We have not been able to find studies that perform speaker attribution under equal circumstances. A study on cabinet meetings (Ruppenhofer et al., 2010) had the goal of annotating all sentences of cabinet protocols with its speaker. They used a rule-based approach exploiting properties of German morphology. Speaker attribution of sentences has also been studied on dialogues in literature, where again the task is to annotate sentences or utterances with speaker information, where this is not explicit. An example is He et al. (2013) who applied supervised machine learning to the task. We do the same in this study but the genre is different and our data points are usually much longer than a single sentence.

Still, the task has similarities with closed-class author attribution. A taxonomy of six feature categories has been proposed for this task by Stamatos (2009): character, lexical, syntactic, structural, semantic, and application-specific. The first two types have the advantage that they can be computed with very little analysis of the text; they include frequency counts of function words, punctuation marks, and short ngrams. Syntactic features can refer to part-of-speech tags or ngrams of these. Structural features include word length and sentence length as well as layout features.

Features requiring detailed analysis of texts, such as full syntactic parsing and topic modelling have also been used. Zhang et al. (2014) used dependency parsing as well as morphological and syntactic features, while Savoy (2013) employed topic modelling as a basis for feature selection. Seroussi et al. (2014) showed how variants of topic modelling can be used to predict authorship and Sari (2018) used topic modelling to analyse which features are effective under different conditions, showing content-based features to be more effective when the diversity of topics in the document set is more varied.

Given a set of selected features that can be used for profiling documents as well as authors, a method is needed to decide among the authors for a given document. Well-known methods based on a selection of frequent words are Chi-Square distance (Grieve, 2007), Burrows's Delta (Burrows, 2002), and Kullback-Leibler Distance (Zhao and Zobel, 2007). All of these compute a distance metric where the author model with the smallest distance to the document model is proposed as the most likely author. Among machine learning methods k-nearest neighbours and support vector machines have been tried, often with good results.

Neural methods have also been applied, sometimes with mixed results. The best overall system at the PAN-2015 author identification task was a character-level RNN language model (Bagnall, 2015), while the neural systems at the cross-domain author identification task at PAN-2018 did not compete well (Kestemont et al., 2018). Most systems at that event used SVMs while the best system was an ensemble system, combining features of three kinds with logistic regression.

More recently, there have been a few examples of author attribution in which the Transformer architecture (Vaswani et al., 2017), which does automatic feature extraction, has been utilised. For example, Fabien et al. (2020) introduced BertAA, a fine-tuned BERT language model for authorship classification. In experiments, the pre-trained model was fine-tuned on three different datasets in the domains of emails, blogs and movie reviews, respectively. State-of-the-art performance was obtained on all three datasets either with plain BertAA or with BertAA with additional features.

## 3 Data and preprocessing

The data collected at The Riksbank have two main sections: minutes from monetary policy meetings and public speeches given by Executive Board members. The minutes are from two periods: One batch starting in February 2000 and ending in May 2007, and another beginning in June 2007 and ending in April 2018. Minutes from the earlier period are truly anonymous, while the minutes from the later period have been anonymised for the purposes of this study. An overview of the data can be found in Table 2.

The speeches have been collected during a somewhat longer period, from 1997 forward. The speeches mostly address the current economic situation and are addressed to a variety of audiences such as banks, regional authorities, chambers of commerce, and parliamentarians.

Both minutes and speeches were originally in

either doc- or PDF-format. Texts were extracted from the PDF-files using the Apache Tika parser[2] accessed via a Python port[3]. From the minutes we then used regular expressions to remove data that was not text such as multiple empty lines, page headers, pagination and table cell data.

The outline of the minutes has changed over the years but is typically divided into four numbered sections. Some minutes have less than four sections and a few of them have more. Each section is supplied with a heading that starts with an initial '§'-sign. The contributions are found in a separate section with a heading such as *Penningpolitisk diskussion*, 'Discussion on monetary policies' or just *Diskussion*. This section is the one from which we extract contributions for the experiments.

A contribution from a board member in the minutes is as a rule introduced with the member's title, e.g., *Förste vice riksbankchef*, 'First Deputy Governor' and full name. All text following this introductory phrase and lasting until a new introduction of the same type is encountered has been allocated to a single contribution. A member may speak at a meeting on several occasions and so we have also collected these together as aggregated contributions. The total number of individual contributions is 900, and the aggregated contributions amount to 385.

| Data type | Numbers |
|---|---|
| Speeches | 399 |
| Meetings / Minutes | 65 |
| Members present at meetings | 5-6 |
| Members during 2007-2018 | 12 |
| Individual contributions | 900 |
|     min length (in tokens) | 11 |
|     max length (in tokens) | 2760 |
| Aggregated contributions | 385 |
|     min length (in tokens) | 68 |
|     max length (in tokens) | 5095 |
| Individual contributions (BERT) | 1738 |
|     min length (in tokens) | 13 |
|     max length (in tokens) | 512 |
| Aggregated contributions (BERT) | 1434 |
|     min length (in tokens) | 32 |
|     max length (in tokens) | 512 |

Table 2: Overview of the data used in the study.

In most meetings six members including the Governor are present. There are a few meetings with fewer members present. It does not happen that a member does not contribute to the discussion at all. Some members have been present at the

majority of meetings, others at only a few e.g., because their period as director ended. The minutes and the speeches have all been parsed by the Sparv parser (Borin et al., 2016). The information obtained from Sparv includes lemmas, part-of-speech tags and word senses, which we have used in subsequent processing.

The speeches, all in edited written form, are known to be given by certain members. All text of a speech, with the exception of some metadata information supplied in the header, has been kept. The main processing of the speeches is word based (frequency counts, topic modelling) and for this reason, we did not clean them to the same extent as the minutes.

For fine-tuning the pre-trained BERT model, we used the raw texts from the minutes as data (the speeches were not used in this setting), masking titles, names and gendered pronouns. The masking was done assuming such information could steer the model towards certain predictions, trivialising the task and hampering generalisation to the truly anonymous setting where this information is absent.

For both the individual and the aggregated data, the length of the contributions varies significantly. As seen in Table 2, the aggregated contributions range from 68 to 5095 tokens. Due to the limitations of BERT handling long text sequences, this posed a problem. Other architectures, such as the Longformer (Beltagy et al., 2020), have been proposed to mitigate this problem. However, in Swedish, BERT is currently the best option. What we did in our experiments, was to chunk the long texts into several smaller texts. This was done by adding up sentences of a text until the addition of one more sentence would yield a text with more than 512 tokens.

## 4 Features used in the experiments

We have framed our problem as a closed set classification task and applied a number of different methods. Burrows' Delta uses lexical features, which are detailed below, in Section 5.1, and the BERT model uses its own feature selection. However, for the SVM and MLP models, we have investigated various properties with the potential to differentiate between members. For each of the properties, one or more features were defined. The focus is on properties and features that relate to content and application.

[2] https://tika.apache.org/
[3] https://github.com/chrismattmann/tika-python

In the rest of this section we motivate the choice of features.

## 4.1 Topic modelling

We assume that the topics members address in their speeches are more or less the same as those they address in meetings as they have different backgrounds, affiliations and areas of expertise. We used lemmatized content words for the topic modelling, where we defined a content word as a word with one of the part-of-speech tags adjective, adverb, foreign word, noun, proper noun, and verb as decided by the Sparv parser. Further filtering was made by applying a frequency threshold and a threshold for spread. We trained multiple topic models with different hyperparameters, we used the NPMI coherence measure (Röder et al., 2015) that estimates coherence among word pairs in a topic based on their pairwise associations, as guide to the final topic model.

After a number of trials we found that the full data set of speeches could best be captured by eleven topics. Each topic constitutes a feature of its own. As a form of evaluation, we asked two researchers at the Riksbank to suggest short descriptions of the topics, based on the ten most probable terms for each topic. Although a few of the topics were more difficult than the others to describe convincingly, they ended up with reasonable descriptions for all of them, shown in Table 3. The fact that the topics are varied and interpretable suggests to us that the model has merits.

## 4.2 Sentiment analysis

Some members may have an overall negative outlook on the economy and/or the proposals discussed in board meetings, while others have a more positive one. We capture this aspect via sentiment analysis, where sentiments from the speeches are compared to sentiments expressed at meetings.

For sentiment analysis we have used a Swedish version of Vader[4] (Hutto and Gilbert, 2014) that also considers a word's sense. Vader is a lexicon and rule-based sentiment analyser. The lexicon in English Vader comprises 5500 lexical entries with sentiment scores between +5 and -5. We used the Swedish SenSALDO 0.2 sentiment lexicon (Rouces et al., 2019) with sentiment scores -1, 0 and +1, that comprises 12287 lexical entries of which 8893 are unique words. It has an accuracy of

---

[4]https://github.com/cjhutto/vaderSentiment

0.89 (Rouces et al., 2019). Word sense disambiguation with the SenSALDO 0.2 lexicon is achieved using the Sparv parsed texts.

Vader also uses booster words, such as *amazingly*, to further refine the sentiment analysis. The booster dictionary used in our analyses is a slightly enhanced version of the Swedish dictionary used for sentiment analysis of consumer support e-mail conversations and comprises 89 items (Borg and Boldt, 2020). That version of Vader uses a smaller lexicon, the Swedish sentiment lexicon (Nusko et al., 2016). It was evaluated showing an 88% correspondence with human annotators.

Vader produces a compound score for each sentence, by summing the valence scores of the words according to their identified sense and normalise this sum to be between -1 (most negative) and +1 (most positive). This gives one feature. We also calculated the amount of positive, negative or neutral sentences yielding another three features. For this, we use the recommendations that a sentence has positive sentiment if the compound score is $\geq 0.05$, neutral if the compound score is between -0.05 and 0.05 and negative if it is $\leq -0.05$[4].

## 4.3 Application-specific features

Some members use more words than others. We capture this aspect by counting the number of words that each member uses, and by computing a member's share of words at a meeting. The relative share of a member's contribution gives a single feature. We also assume that the speaking order that is reported in the minutes reflects the actual speaking order at the meeting. If this order is dependent on the board member's status, or role, it could be fairly stable over time, or only change gradually. This aspect gives rise to six features corresponding to being the first speaker, the second speaker, and so on.

It is known for each member whether they have entered a reservation against the majority decision. We assume that members may differ in their incidence of entering reservations. The probability of entering a reservation is used as a feature.

## 4.4 Feature selection

Table 4 shows the properties we have investigated. Topic distribution and Sentiments cover the contents of contributions while the rest are application-specific capturing aspects of members' meeting behaviour. For each property, we first determined whether it could have some predictive value on its

| Topic | Description | Most probable terms |
|-------|-------------|---------------------|
| 0 | Monetary policies general | *styrränta, inflationsförväntning, inflationspolitik, mena, nominell*
policy rate, expectation on inflation, inflation targeting, mean, nominal |
| 1 | Housing and private debt | *skuldsättning, skuld, bostadspris, bostad, bostadsmarknad*
indebtness, debt, price of housing, housing, housing market |
| 2 | Financial stability and macro prudential | *myndighet, verktyg, institut, makrotillsyn, regelverk*
public authority, tools, institute, macro supervision, regulations |
| 3 | Public debt and quantitative easing | *balansräkning, obligation, statsobligation, avkastning, miljard*
balance sheet, bond, gobernment bond, returns, billion |
| 4 | Transparency and communication | *direktion, möte, öppenhet, prisnivå, kommunikation*
Executive board, meeting, transparency, price level, communication |
| 5 | Labour market | *arbetsmarknad, produktivitet, vänta, inflationsförväntning, inflationsrapport*
labour market, productivity, wait, expectation on inflation, inflation report |
| 6 | Monetary policy general II | *tillgångspris, resursutnyttjande, inflationsmålspolitik, mena, nominell*
asset price, resource utilization, inflation targeting, mean, nominal |
| 7 | International trade euro area | *euro, eu, emu, konkurrens, handel*
Euro, EU, EMU, competition, trade |
| 8 | International trade general | *offentlig, sparande, diagram, bytesbalans, export*
public, savings, diagram, balance of payments, export |
| 9 | Payment system | *betalning, pengar, kontanter, betalningssystem, infrastruktur*
payment, money, cash, payment system, infrastructure |
| 10 | Inflation targeting and the policy rate path | *resursutnyttjande, diagram, räntebana, stabilisera, hållbar*
resource utilization, diagram, policy rate path, stabilize, sustainable |

Table 3: Descriptions of the produced topics with the five most probable terms.

own using both MLP- and SVM-systems[5]. It can be seen from Table 4 that all selected properties give performance above a random baseline which, for six participants present in each meeting, would give a theoretical accuracy of 16.7%. Topic distribution is by far the property that has the best results.

In total, our feature set consists of 37 features. Since we are interested in how these features impact member classification, we employed two different feature selection methods. The first approach is a Recursive Feature Elimination (RFE) which is able to find a set of features that carry the most predictive power. The second is based on a Python implementation[6] of the Boruta algorithm (Kursa and Rudnicki, 2010). The rationale behind using Boruta is the algorithm's ability to provide a set of relevant features, contrary to the minimal optimal feature sets provided by for example RFE. This means that we with Boruta are able to get a set of all features that have some impact on the prediction, while with RFE we can choose to extract the N most important features. By using a combination of these algorithms, we can therefore gain knowledge about which features carry the most predictive power if we wanted to slim down the classification model, but also a picture of which of the features provide at least some information for the classification task.

# 5 Experiments

This section elaborates on the details of the different systems and their performance. All results are shown in Table 5.

## 5.1 A traditional system: Burrows' Delta

For comparison, we tested an implementation of Burrows' Delta under different conditions. Three different feature sets were used, one relying solely on the most frequent words in the corpus of speeches, another where proper nouns were removed, as these include references to the speaker we wish to identify, and a third relying on the most frequent content words, where a content word was defined as a noun, verb or adjective. Following Evert et al. (2015) we also looked at the effect of normalising the feature vectors and compared two different measures: Manhattan distance and Cosine similarity.

Initial tests were made on a corpus where all contributions from one member had been collected into one text yielding a total of 12 texts. These suggested that the frequency-based features gave slightly better results than the other two, with 7 out of 12 members being predicted correctly, and 9 out of 12 being included in the two first predictions. This selection of features was then used for predicting the speaker of contributions at meetings. The number of features was also varied showing clear

---

[5]See section 5.2 for a description of the experimental setup

[6]https://github.com/scikit-learn-contrib/boruta_py

| Property | Features | Accuracy (SVM) | Accuracy (MLP) |
|---|---|---|---|
| Length (absolute) | 1 | 30.57% | 26.67% |
| Length (relative) | 1 | 23.19% | 28.33% |
| Order (only position) | 1 | 25.13% | 21.25% |
| Order (probabilities) | 6 | 42.31% | 40.97% |
| Reservation | 1 | 23.10% | 21.25% |
| Sentiments (compound) | 1 | 18.77% | 16.45% |
| Sentiments (ratios) | 3 | 23.11% | 16.06% |
| Topic distribution | 11 | 63.70% | 62.84% |
| Burrows Delta | 12 | 24.86% | 29.34% |

Table 4: Properties used and their performance as single predictors for the SVM and MLP models.

improvements from 300 features upwards with a peak around 500. Using normalised feature vectors and cosine distance consistently gave better performance by two or more points. The best results are reported in Table 5.

We observe that the best result for the aggregated contributions is close to that for the topic models. We also see that results drop when predicting speakers of individual contributions but are still far above chance. Adding more features does not generally improve predictions.

We can also note that the performance of using the Burrows' Delta models for different speakers to generate features to be included in an SVM-classifier differs greatly from using the standalone system for classifying members with Burrows' Delta.

## 5.2 The SVM and MLP systems

Each type of feature was first tested individually to see whether it could beat a random baseline. The results are reported in Table 4.

The systems, written in Python, use the scikit-learn library (Pedregosa et al., 2011), with the implementations of support vector machines (SVC) and multilayer perceptrons (MLPClassifier) as the algorithms for the classification task.

We used a 5-fold cross-validation procedure to randomly split the data into training and test data. Since we wanted to do the prediction of the contributing members on a meeting level, we let the individual meetings be the unit assigned to either the train or test portion of each fold, with all member contributions extracted from the particular meeting. In the cross-validation procedure, it is however customary to balance the classes (in this case, the members) evenly across all folds, but as a consequence of the importance of keeping the integrity of each meeting, it was not possible to achieve a perfect balance of classes in all folds.

In each training fold, we performed a second 5-

fold cross-validation procedure to optimise the hyperparameters of the selected classification model. For the SVM, we optimised the C and gamma values with a radial basis function (RBF) kernel. The MLP was optimised with its hidden layer sizes and the L2-regularisation term (named alpha in scikit-learn) for the *Limited Memory Broyden–Fletcher–Goldfarb–Shanno* (lbfgs) solver.

We implemented a custom prediction step with two restrictions for the classification task, namely, for each meeting;

- Only members present in the meeting can be predicted.
- A member can only be predicted once per meeting.

## 5.3 Ensemble systems

Using the same features and the two restrictions just described, two ensemble systems of SVMs and MLPs were implemented[7]. The first is a soft voting system, where both an SVM and an MLP are trained as described in section 5.2. At the prediction step the classification probabilities, for each possible board member, of both the SVM and MLP are added together and averaged between the two classifiers. The board member with the highest average probability is then subsequently selected as the classifier output for the given meeting. Since we noticed subtle differences in how the SVM and MLP predicted certain meetings, the rationale behind this approach was to try to make a more robust prediction, leveraging the strengths of both classifiers.

The second ensemble system is a hybrid of an MLP and an SVM, following the method used in Garg et al. (2021). The system consists of an MLP that is trained on the training splits in a regular fashion, but whose weight matrix from the final

---

[7]The cross-validation and hyperparameter optimisation were performed in the same fashion as described in section 5.2

hidden layer is used as features by an additional SVM classifier.

## 5.4 BERT-based system

As with the SVM and MLP systems, we used a 5-fold cross-validation procedure to randomly split the data into training and test data. The fine-tuning procedure was implemented using Transformers (Wolf et al., 2020) and PyTorch (Paszke et al., 2019). To make this method comparable with the other methods described, we combined the predictions for smaller chunks of a given contribution into one single prediction. Thus, we had to keep track of the contribution ID when splitting into training and test data and make sure that all smaller chunks for a given contribution were in the same partition. The combining was then done by summing the raw output scores from the model for all chunks of a given contribution before picking the class with the highest score as the prediction. This way, we got a single prediction for each contribution.

For both the aggregated and individual data, we did experiments of two kinds, one where only the members present at a particular meeting were considered when aggregating predictions and one that disregarded the notion of meetings. The former setting is similar to the setting used for the SVM and MLP systems, with the only difference being that each member in a meeting could now be predicted multiple times. In the latter case, no information about what members participated at a particular meeting was given to the model. Thus, the model had to predict the member from the pool of all 12 members. Surprisingly, at the end of training for each fold, the results were exactly the same in all cases but one where the first approach had an increase in accuracy of approximately 1% compared to the second approach. This effect was seen in both the aggregated and individual data.

In each fold, the data was prepared as input to the BERT model by retrieving input ids, and the attentions mask for each batch of sequences. A batch size of 8 was used, and the model was fine-tuned for 10 epochs on a Tesla P100-PCIE-16GB GPU with a learning rate of $10^{-5}$. 10 epochs seemed to be suitable for this problem and dataset, as loss converged without causing overfitting.

## 6 Results

The results for classification accuracy of all tested systems are presented in Table 5. For all systems,

| System | Contributions | |
| --- | --- | --- |
| | Aggregated | Individual |
| Delta, standard, 500feats | 55.54% | 33.89% |
| Delta, normalised, 500feats | 60.33% | 42.41% |
| SVM (*RFE*) | 78.20% | 54.18% |
| SVM (*Boruta*) | 79.70% | 57.55% |
| SVM (*All*) | 78.56% | 56.98% |
| MLP (*RFE*) | 76.22% | 52.45% |
| MLP (*Boruta*) | 77.15% | 54.14% |
| MLP (*All*) | 74.66% | 54.55% |
| Soft voting (*RFE*) | 77.25% | 54.50% |
| Soft voting (*Boruta*) | 77.95% | 55.83% |
| Soft voting (*All*) | 76.48% | 57.52% |
| Hybrid (*RFE*) | 78.71% | 54.78% |
| Hybrid (*Boruta*) | 78.50% | 54.66% |
| Hybrid (*All*) | 79.31% | 55.70% |
| BERT | 94.81% | 83.78% |

Table 5: Results of the classification accuracy for different systems, feature sets and types of member contributions.

as expected, the aggregated contributions score higher than the individual contributions. Classifying aggregated (and longer) contributions are naturally a less complex problem compared to individual contributions due to the reduced number of classifications to be performed per meeting. It should however be noted that the tested systems, especially the system based on BERT, are able to handle this change of scope in an acceptable manner considering the increased task complexity.

Furthermore, the results indicate that the SVM and MLP classification methods performed significantly better than the random baseline and that the differences between these methods were relatively small. When including all the features listed in Table 4, we saw a lower classification performance for all feature-based systems, compared to when we included only a subset of the features (see Table 5). The best results were generally found with the subset of features selected by the Boruta-algorithm, referred to as *Boruta* in Table 5. The best performance of any feature based system can be seen in the standalone SVM system with an accuracy of 79.70% on the aggregated data and boruta feature selection. The best performance of the Ensemble systems were found in the hybrid system with an accuracy of 79.31%, followed by the soft voting system with an accuracy of 77.95%. The standalone MLP system performed the generally lowest scores, with the highest being 77.15%,

An even smaller subset of features (the *RFE* feature subset), including the 10 features with the most predictive power according to the Recursive Feature Elimination, were able to perform almost on

par with the other feature sets. This also aligns with what was seen when each feature was tested individually (see Table 4), where some of the features scored close to the random baseline. The features present in the subset created with *RFE*, **topics**, **length (absolute)**, and **speaking order**, were also some of the highest performing individual features. It should however be noted that not all of the topics are included among the top 10. Three of the topics are not (topic 6, topic 7, and topic 9). These topics were also some of the few features that most often were omitted as features by the Boruta algorithm. All this taken into consideration, we can conclude that a small number of features carry great predictive power for the classification task.

The fine-tuned BERT model obtained 94.81% accuracy for all folds combined on the aggregated data, and an accuracy of 83.78% on the individual data. Since the data used for the BERT model had to be split into smaller chunks to fit the input limit of BERT, we tried the same chunking approach for all the non-BERT systems. Since some of the aggregated contributions were fairly long (see Table 2) the total number of contributions increased significantly, while also rendering the restriction of only being able to predict a member once per meeting less effective. The SVM, MLP, and ensemble systems did therefore perform worse with this data chunking approach, resulting in accuracies between 55-58% on the aggregated data.

## 7 Conclusions and discussion

In this work, the main purpose has been to investigate a set of interpretable features for identifying speakers from minutes. With the aid of feature selection algorithms, we are able to pin down the most important features, while also excluding some of the less relevant, and simultaneously improve the classification performance. Topic models generated from speeches given by directors of the board of The Riksbank turned out to be a good predictor of what they say in board meetings. Combined with other features such as wordiness, speaking order and sentiment analysis we could reach an accuracy close to 80% in predicting which director said what. Not surprisingly the fine-tuned BERT model has the best performance in predicting which board member made a certain contribution. This is in line with the performance of similar models in other attribution tasks (cf. Fabien et al. (2020)) and points

to Transformer-based models being good feature extractors. While we have only investigated one corpus of minutes, the methods we've tried have a wider application; similar types of meetings and minutes are common in financial and other public institutions where transparency and accountability is an issue.

The success of the BERT-model suggests that members are consistent in their argumentation across meetings. An interesting aspect is the fact that the minutes of the meetings are not written by the members, which should make this task harder than standard author attribution. Given this, we find that the BERT model provides a strong benchmark for de-anonymisation of minutes.

The BERT-model, unlike the features used for the other models, is not easily interpretable. Yet, as new techniques for interpreting models such as BERT are emerging, we would like to investigate what the BERT-model actually considers when making predictions. For example, whether it looks at stylistic features in how the minute taker writes about a particular member or features more related to the content and topics of the contributions.

The properties, coupled with analysis of the overall differences of the minutes under the two conditions, are likely to be helpful in future research on de-anonymising the minutes from the earlier period. Although members are not referred to by name there is a similar structure to the minutes and the discussions so that contributions can be identified. While performance may be lower for all models when applied to minutes for the earlier period, the data obtained from the non-anonymous minutes could then be used for training. For example, we know from confusion matrices which speaker models are often confused.

There are features we have not yet investigated such as members' style of argumentation, or rhetorical structure, which potentially could be helpful. As we now also have identified the topics discussed during the meetings, we can analyse members' attitudes, i.e. sentiments, towards each topic. This can also be included in our model. An analysis of the parse trees of contributions could also yield features at a more fine-grained level than topics, such as individual members' hobby-horses.

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
