# OpenReview forum: "Who said what? Speaker Identification from Anonymous Minutes of Meetings"
_NoDaLiDa/2023/Conference — NoDaLiDa 2023_

### Official Review · Reviewer_vzAH · 2023-03-05
**A study of various methods for speaker identification in Swedish board meeting transcripts and/or a de-anonymization attempt**

**Rating:** 7
**Confidence:** 4

**Review:**

A very interesting read on different methods used to identify speakers of Swedish board meetings. This can also be viewed as a de-identification attempt where the task is to re-identify anonymized data. Although the paper was well written and all the definitions, data, and methods clearly explained, the difference in performance between the methods suggested and a simple BERT model is quite high. It was interesting to see how different sets of features affected each method, but I was wondering whether more effort should have been made to experiment with just a BERT model more (as also mentioned in the Conclusion part)

Pros:
1. Clearly explained dataset choice and features
2. A significant number of experiments both on features for each method and on methods used

Cons:
1. I don't believe that the reasoning behind using methods such as a SVM for example was adequately justified, seeing how it is widely accepted that a BERT model typically outperforms such attempts for sequence classification tasks (as also shown in Table 4)

- As a side-note, the authors mention BERT's token limit and how they tackled that. The similar LongFormer model can handle input up to 4096 tokens and can aid in tasks where context is important as this one is.

**Paper Type:**

Long paper

---

### Official Review · Reviewer_MuyL · 2023-03-07
**Who said what? Speaker Identification from Anonymous Minutes of Meetings**

**Rating:** 7
**Confidence:** 5

**Review:**

The paper describes work act to identify speakers at board meetings of Sveriges Riksbank from the meetings’ minutes and the speeches of the speakers. Many techniques, features and experiments are applied and described. The best results for the task were achieved by a transformer with a BERT-based model.
The subject is relevant for NODALIDA.  I had some difficulties in keeping track of the many techniques, features, experiments and the motivation for applying them because feature extraction and models are in some cases connected, while their description is in different sections.
Since the paper concerns so many experiments and steps, some details are not given, and therefore some of the experiments cannot be replicated.
In the following I go through some of the points that should be clarified:
-	I guess that “speeches” (from line 239) refers to the transcriptions of the speeches (pdf format on line 255). Are these transcriptions literal?
-	What does “noise” mean in this context (line 318)? This question is related to the previous one.
-	Line 258: using open-source software – which?
-	Line 354:  in an external corpus such as Wikipedia – was the corpus used Wikipedia or not? If not which one?
-	Line 390: we ranked different topic models – Which models? If there is no space just write which model you have used
-	Line 557: Why are proper names removed in one of the models?
-	Through the paper the term “Behavioural features” is used and it is confusing since the features are not what is usually known as behavioural in speeches  (e.g. pauses, dialogue marks,  auto-corrections etc.) – Call your features something else (other features?)
-	The comparison of results in table 4 shows that BERT gives significantly better results than all other techniques.  However, BERT vs. the other techniques is only shortly discussed while it is crucial trying to explain why it works better (some of them are clear knowing how BERT works)
-	You use a different way to split data when evaluating the results of BERT. What is the motivation?
Some minor points:
-	Line 036 “of thereof” - -> “thereof”
-	Line 529 a feature set that is all relevant  - -> a set of relevant features
-	Line 570 – to what does “the other two” refer
-	Links to the various tools used should be added in footnotes or appendices
I suggest that the paper is restructured so that the features used in the experiments with SVM and MLP are presented together with these experiments since they are only used there. Also I would remove some irrelevant details, while explaining clearly what you do.

**Paper Type:**

Long paper

---

### Official Review · Reviewer_JNPn · 2023-03-09
**Person identification from Swedish Central Bank meeting minutes.**

**Rating:** 5
**Confidence:** 5

**Review:**


The problem of person identification discussed and experimented in this paper is extremely interesting. However, the data and the exact task are not explained well enough to be able to relate the current work with what is discussed about related work in section 2. This could be a great paper if the data, the task and the evaluation setting would be clearly defined and the information about some extra experiments with no consequences to the results or conclusions would be left out. Now they confuse the reader even more.

Pros:
- the problem is very interesting and it has not been done at least on Swedish language before

Cons:
- the data is not explained well enough
- the task is not defined well enough
- the evaluation setting is not defined well enough
- due to the three previous points, it is not possible to evaluate the results and the conclusions of the paper

Detailed comments:

Before reading the related work, or in its beginning, I would like to have known how do the meeting minutes in question relate to what was actually said in the meeting. Are they interpretations/summaries written by a secretary or are they direct quotes of what was said in the meeting. If they are direct quotes, whether the utterances are normalized or not? As of the beginning of the related work, I'm unaware whether the task is just opinion detection or also idiolect detection? "reports of oral contributions" could be anything. Maybe the data should be discussed more before the related work? Preprocessing of the data could still come after, but to relate to and make the related work discussion more interesting and relevant, I would like to understand the task at hand better.

017: "come" -> "comes"

036: "part of thereof" -> "part thereof"

Section 3: As this is not exactly a dataset paper I suggest describing only features that are actually used in the experiments. So attendance records and voting behaviour is not necessary to mention (except that later it can be seen that you maybe use the attendance records, but maybe not voting behaviour). You mention that you got the data in pdf-format. From where? With what kind of license?

256-257: Which open-source software? Give credit when you can if you have space. (link in footnote would be enough I guess).

306: While it is kind of self-evident to us reading this now, it might be good to mention somewhere in the beginning that we are talking about minutes written in the Swedish language.

317-318: "we did not find it worthwhile to remove noise.": What noise? Why is word based processing a reason not to remove this noise?

320-321: Did you mask titles and names also inside the speeches? How do gendered pronouns trivialise the task? How do you know they will trivialise the task?

326-327: We need a reference to somewhere where the limitation of BERT handling long sequences is demonstrated and explained.

332: Where does this threshold come from?

349: I don't know what "application" here means?

Section 4: Table 2. At this point, the reader has no idea how the accuracies shown in the table are calculated? Are you choosing your properties by evaluating them on the test data? Also, the reader does not know at all what are some of these properties like "Sentiments" at this point. Maybe this table should come after explaining these properties better?

357: You mention MLP here, but only show SVM accuracy in the table?

359-361: What six participants? Weren't there 12 persons to choose from? It seems the reader has no idea what is the task at hand.

Table 3: "styrränta" is not in italics similarly to the other Swedish words.

512: What does "reservation" mean? You should go over the dataset in detail as early as possible and explain what is there to use.

604: What was the minimum unit in splitting the data? E.g. is the training data in-domain (from the same meeting) with the test data?

626-632: You should clarify the experimental setup before the experiment results. Now the reader does not know whether these two restrictions apply to only experiments in section 5.2.

655: Instructions for NoDaLiDa 2021 Proceedings, section 3.6: "refrain from using full citations as sentence constituents."

666-667: These are the only times Huggingface and PyTorch are mentioned. Links or references are needed.

681-687: As the experimental setup was confusing already before this, it is not possible for the reader to figure out how these changes relate to what was before. All these possible configurations cast also doubt in whether the results with different methods are at all comparable with each other.

Table 4: Are these results comparable to each other? Even after going back and forth the article, I have no idea what to make of these results. Especially interesting would be to know whether the training material can have contribution from one director in the same meetings as is in the test set?

786: Why do you think this loss of performance happens?

803: How does this 75% compare to the almost 95% attained by BERT in Table 4? 95% is not "well above" 75%.

**Paper Type:**

Long paper

---

### Decision · Program_Chairs · 2023-03-17

Accept